# Brief intervention to reduce fatigue impact in patients with inflammatory arthritis: design and outcomes of a single-arm feasibility study

Emma Dures ,[1,2] Susan Bridgewater,[1,2] Bryan Abbott,[1] Jo Adams ,[3] Alice Berry,[1,2] Lance M McCracken,[4] Siobhan Creanor,[5] Sarah Hewlett ,[2] Joe Lomax,[6] Mwidimi Ndosi,[1,2] Joanna Thorn ,[7] Marie Urban,[1] Paul Ewings[8]

For numbered affiliations see end of article.

**Correspondence to**
Dr Emma Dures;
Emma2.Dures@uwe.ac.uk

## ABSTRACT

**Objectives** Patients with inflammatory arthritis report that fatigue is challenging to manage. We developed a manualised, one-to-one, cognitive–behavioural intervention, delivered by rheumatology health professionals (RHPs). The Fatigue - Reducing its Effects through individualised support Episodes in Inflammatory Arthritis (FREE-IA) study tested the feasibility of RHP training, intervention delivery and outcome collection ahead of a potential trial of clinical and cost-effectiveness.

**Methods** In this single-arm feasibility study, eligible patients were ≥18 years, had a clinician-confirmed diagnosis of an inflammatory arthritis and scored ≥6/10 on the Bristol Rheumatoid Arthritis Fatigue (BRAF) Numerical Rating Scale (NRS) Fatigue Effect. Following training, RHPs delivered two to four sessions to participants. Baseline data were collected before the first session (T0) and outcomes at 6 weeks (T1) and 6 months (T2). The proposed primary outcome was fatigue impact (BRAF NRS Fatigue Effect). Secondary outcomes included fatigue severity and coping, disease impact and disability, and measures of therapeutic mechanism (self-efficacy and confidence to manage health).

**Results** Eight RHPs at five hospitals delivered 113 sessions to 46 participants. Of a potential 138 primary and secondary outcome responses at T0, T1 and T2, there were 13 (9.4%) and 27 (19.6%) missing primary and secondary outcome responses, respectively. Results indicated improvements in all measures except disability, at either T1 or T2, or both.

**Conclusions** This study showed it was feasible to deliver the intervention, including training RHPs, and recruit and follow-up participants with high retention. While there was no control group, observed within-group improvements suggest potential promise of the intervention and support for a definitive trial to test effectiveness.

## INTRODUCTION

Inflammatory arthritis (IA) is a group of multisystemic, autoimmune conditions characterised by pain, joint swelling and stiffness, and fatigue. The most common of these conditions is rheumatoid arthritis (RA).[1] It is

## STRENGTHS AND LIMITATIONS OF THIS STUDY

⇒ This feasibility study has established that rheumatology health professionals can train and deliver a brief, low-cost intervention for fatigue in inflammatory arthritis.

⇒ The low levels of attrition and high levels of data completeness suggest the outcomes collected are appropriate for a definitive trial.

⇒ Within-group improvements were observed, although this could have arisen from regression to the mean or the small sample size.

⇒ The lack of a control arm means that the feasibility/acceptability of randomisation has not been tested.

estimated that over 750 000 adults in the UK have an IA.[2 3] Challenges for patients with IA include unpredictable fluctuations in symptoms, functional disability and managing complex medication regimens.[4] Treatment options include pharmacological, non-pharmacological and surgical interventions to control symptoms, prevent joint damage and improve mobility and function.[5] In the UK to date, treatment for IA is typically provided in secondary care by multidisciplinary rheumatology health professionals (RHPs), including physicians, nurse specialists, occupational therapists and physiotherapists.

Although the clinical manifestations vary, fatigue is a prevalent and often disabling symptom across types of IA[6–8] It is experienced by patients as a fluctuating, unpredictable symptom that impacts on all aspects of daily life.[9] An international study of >6000 IA patients found that one out of every two was severely fatigued, defined as scoring ≤35 on the Short-Form (SF-36) Health Survey Vitality Scale.[10] Despite the high prevalence and impact of the symptom, patients perceive that often their fatigue is not addressed in rheumatology consultations.[11] UK research

with >1200 IA patients found that 82% wanted support to manage the impact of pain and fatigue.[12] RHPs have reported that they recognise that fatigue is an issue for patients, but there is a lack of evidence-based resources that they can use in clinical practice.[13]

Fatigue is a complex, multifaceted phenomenon, the mechanisms of which are not fully understood. Challenges include the difficulty of measuring fatigue, and the high number of previous studies that have used cross-sectional designs, making it hard to understand directionality and attribute causality.[6] However, from the evidence available, fatigue in IA is associated with inflammation, pain, disability, sleep, depression and health beliefs, implying complex, multicausal pathways.[14] A systematic review found that biological treatments in patients with active RA can lead to a small to moderate improvement in their fatigue, suggesting that optimal disease activity management should be part of fatigue management.[15] However, biological treatments are not prescribed for IA-related fatigue, and there is evidence that patients can experience fatigue during remission.[16] A systematic review for non-pharmacological interventions concluded that physical activity and psychosocial interventions, including cognitive–behavioural therapy (CBT), provide benefit in relation to self-reported fatigue in adults with RA.[17] This evidence has underpinned several CBT-based self-management interventions for fatigue.[18 19] Although clinically effective, they are highly structured, stand-alone interventions comprising at least six patient contact sessions. Consequently, they are time-consuming for patients to attend and for RHPs to deliver.

In response, we developed the Fatigue - Reducing its Effects through individualised support Episodes in Inflammatory Arthritis (FREE-IA) study. As part of the study, we designed a brief, one-to-one intervention that aims to reduce fatigue impact by supporting patients to identify the thoughts, feelings and behaviours perpetuating their fatigue online supplemental summary 1). Patients can then use this understanding as the basis for making adaptive behaviour changes and enhancing their coping skills. The intervention is based on self-determination theory, which addresses motivation and competence to behave in effective and healthy ways; self-efficacy, a belief in one's ability to successfully engage in a course of action; and guided discovery (the 'Ask don't tell' approach rather than didactic information and advice-giving).[20–22] The intervention was designed by a multidisciplinary team from nursing (SH), occupational therapy (JA) and psychology (LMM and ED) and written as a manual, designed to be used after training in cognitive–behavioural approaches, daily dairies and goal setting. It comprises two to four sessions, each designed to last 20–30 min (table 1). The first two sessions are core and designed to take place face to face and within 2 weeks. Up to two additional optional sessions can take place face to face or remotely, for example, by telephone or video, within the subsequent 4 weeks.

Our study design was informed by the Medical Research Council's framework for developing and evaluating complex interventions.[23] Before investing in a definitive randomised controlled trial (RCT) to test an intervention's clinical and cost-effectiveness (evaluation stage), the research team

**Table 1** Overview of intervention structure and content

| Sessions 1–4 | Key topics | Key handouts |
|---|---|---|
| Engagement and validation | Identify fatigue drivers<br>Activity management | Fatigue overview<br>Activity diaries |
| Daily diary, goals, action planning | Boom and bust; avoidance and withdrawal<br>Drainers and energisers | Pacing<br>Goal setting<br>Activity diaries |
| Sleep and rest | Nature of sleep difficulties<br>Sleep myths and strategies | Sleep and relaxation<br>Activity diaries |
| Stress and relaxation | Symptoms of stress<br>Coping resources | Stress bucket<br>Activity diaries |

should have a reasonable expectation that the intervention could have a worthwhile effect, based on existing evidence and theory (development stage). They should also examine whether the evaluation procedures are likely to be deliverable and acceptable (feasibility stage). Researchers are advised to use a mix of quantitative and qualitative methods to resolve the main uncertainties that might impede study delivery. To achieve this, we designed the feasibility study FREE-IA.

Our aims were to:
► Design and deliver intervention training to RHPs.
► Recruit patients to the intervention.
► Determine the completeness of outcome measurement data collection from patients who participated in intervention sessions.
► Identify the optimum approach for a cost-effectiveness evaluation to be conducted alongside a definitive RCT.

We also examined the acceptability of the intervention from the perspectives of patients who participated and RHPs who undertook training and delivery, via telephone interviews. These data are reported separately in a qualitative process evaluation.

## MATERIALS AND METHODS

We used a single-arm feasibility study design comprising three phases:
► Phase 1: delivery of intervention training to RHPs.
► Phase 2: patient recruitment and intervention delivery.
► Phase 3: data collection and analysis.

### Phase I

We developed and delivered intervention training face to face. We included overviews of the IA fatigue evidence base, underpinning psychological theories and materials from the manual (cognitive components); skills demonstrations from the training team (modelling/illustrational component); skills practice using rheumatology-specific vignettes, with observation and feedback from the training team (experiential/behavioural component); and a problem-based learning approach, with RHPs using examples from their clinical practice.[24] Training was designed and delivered by ED, SH and LMM and patient research partners MU and BA.

## Phase II

Individual secondary care sites made local decisions about their optimum strategy to invite patients to participate in the study. Eligibility criteria were rheumatology patients at a participating site; age 18 years and over with a clinician-confirmed diagnosis of IA; with a score ≥6/10 on the BRAF NRS Fatigue Effect[25] and with fatigue that they considered recurrent, frequent and/or persistent; and who were not accessing support for their fatigue at the time of invitation. Patients who were unable to complete questionnaires in English unaided and/or patients lacking capacity to give informed consent were not eligible. Patients interested in participating completed and mailed their screening sheet to the study coordinator SB, who assessed their eligibility for the study. Following confirmation of eligibility, SB mailed a baseline data pack to patients who were interested in taking part. The pack comprised a consent form and a questionnaire to collect demographic and clinical data and the proposed outcome measures to be used in the definitive RCT (see phase III). SB asked patients to complete the baseline data pack, including the consent form, and to bring it to their first face-to-face intervention session.

After training, RHPs delivered intervention sessions to recruited patients. To inform patterns of uptake, amendments to the intervention and the cost of delivery, we asked RHPs to record the number and duration of intervention sessions delivered to each participant and the mode of delivery, for example, face to face, by telephone or by video. Once they had experience of delivery, we asked RHPs to audio-record the intervention sessions, if the participant consented, to assess how the intervention was delivered. We designed a pro forma to guide assessment of competence and fidelity to the intervention. It comprised two parts: (1) inclusion of intervention content/topics and (2) use of facilitative approaches by the RHP. In each section, research fellow AB scored the extent to which planned content was present (0=not present, +=attempted/present, ++=present/a key focus) and made notes to include examples and reflections. This information was for process evaluation purposes (to be reported separately) and not as feedback for the RHPs delivering the intervention.

## Phase III

After baseline (T0), we collected quantitative outcomes data from participants at two time-points: 6 weeks postintervention (T1) and 6 months postintervention (T2). We defined postintervention as 6 weeks after core session 1 because it covered the maximum intended period of exposure to the intervention. Our likely primary outcome in a future RCT is fatigue impact, measured using the BRAF-NRS Fatigue Effect.[25] We also collected validated secondary outcomes:

► BRAF-NRS Fatigue Severity.[25]
► BRAF-NRS Fatigue Coping.[25]
► Rheumatoid Arthritis Impact of Disease[26] (pain, functional disability, fatigue, sleep, coping, physical and emotional well-being).
► BRAF Multidimensional Questionnaire.[25]

► Modified Health Assessment Questionnaire[27] (functional disability).

Measures of therapeutic mechanism:
► The Rheumatoid Arthritis Self-Efficacy Scale[28] (beliefs reflecting confidence in one's capacity to function despite symptoms).
► The Perceived Health Competence Scale[29] (feelings of capability to manage health outcomes).
► The Health Care Climate Questionnaire[30] (perceptions of the extent to which a health professional is autonomy supportive).

SB collected the proposed primary outcome by telephone and the secondary outcomes via an outcome measures pack that was mailed to participants at T1 and T2. Participants were asked to complete the questionnaires and mail them back.

The FREE-IA Project Management Group approved analysis plans for the statistical outcomes and health economics. Methodologists PE, JL and SC conducted analysis of the statistical outcomes. For each self-reported questionnaire, the total scale and subscale scores were calculated in line with published guidance, including the use of imputation for unanswered questions (online supplemental table S2). Outcome scores are reported as means and SD, plus ranges, at each of the three time points. In addition, the mean change from T0 to T2 for each (sub)scale, with 95% CIs, is presented.

Health economic outcomes were analysed by health economist JT. Health-related quality of life (EuroQol-5 Dimensions-5 Levels)[31] was collected at T0, T1 and T2, and valued using the van Hout crosswalk method based on UK population preferences.[32] Mean quality-adjusted life years (QALYs) were calculated over the 6 months of follow-up. A bespoke resource-use questionnaire was developed in consultation with patient partners, covering: (1) NHS and personal social services (PSSs) and (2) patient perspectives. An estimate of the cost of delivering the intervention itself was derived from study records. Standard sources were used to assign unit costs (2019) to each of the resources measured[33–36] and mean usage (eg, appointments), mean costs and SD were calculated over the 6 months of follow-up using all available cases. A non-comparative cost–consequences matrix was constructed.

## Patient and public involvement

The research study, including the question of whether it would be feasible to train RHPs and deliver the intervention, was developed with patient research partners BA and MU, who have experience of living with IA and fatigue. They were coapplicants in the funding application and are coauthors on this manuscript. The proposal was also discussed with the Patient Advisory Group in the Rheumatology Department of the Bristol Royal Infirmary. BA and MU reviewed all patient-facing literature, shaped the bespoke health economics questionnaire, supported delivery of the intervention training, provided additional materials for RHPs

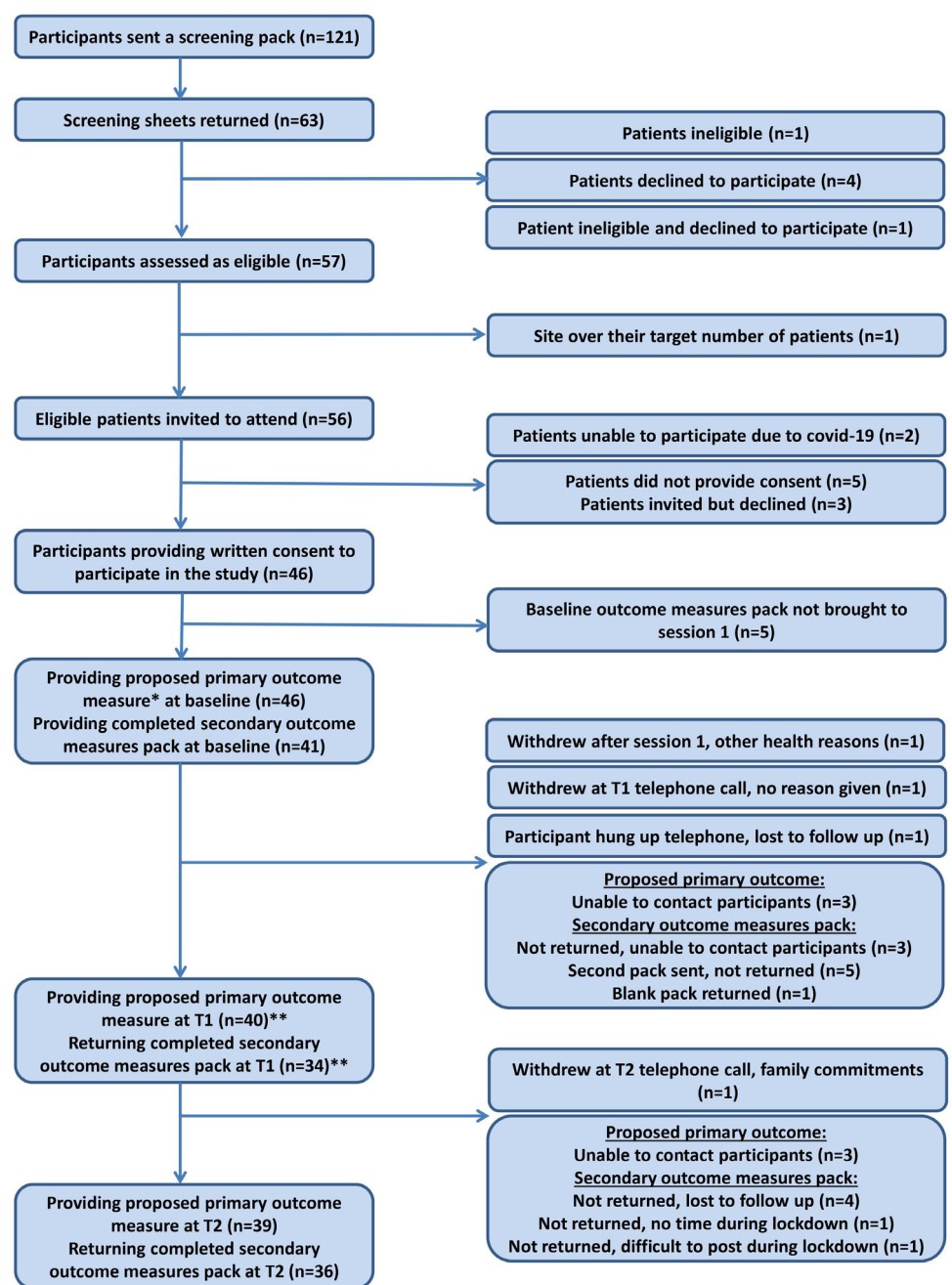

**Figure 1** FREE-IA flow diagram of participants. *The proposed primary outcome was collected by telephone. **Some participants did not return T1 outcomes but remained in the study and subsequently returned T2 outcomes. FREE-IA, Fatigue - Reducing its Effects through individualised support Episodes in Inflammatory Arthritis.

delivering the intervention, advised on recruitment and helped to interpret the study findings. After study completion, they reviewed the written summaries that were sent to study participants, including patients and RHPs who had taken part.

## Analysis and results
### Delivery of intervention training to RHPs

We delivered face-to-face training three times, with different RHPs each time. In total, 12 RHPs (eight nurses, two occupational therapists, one associate rheumatology practitioner and one clinical research practitioner) from

six hospitals attended. The first training took place over 2 days at the hospital where the central study team are based, with seven RHPs from four sites and lasted for approximately 13 hours. Subsequently, one site withdrew from the study after their two RHPs had attended training but before recruiting patients due to logistical challenges of intervention delivery at their hospital. Subsequently, two new sites joined the study, with training delivered over one and a half days (approximately 10 hours) at the same central study team hospital to four RHPs. The third training lasted for 1 day (approximately 5 hours) and was delivered by ED at the hospital of an individual RHP from

**Table 2** FREE-IA participant demographics

|  | Study participants (n=46) |
|---|---|
| **Sex, n (%)** | |
| Male | 9 (19.6) |
| Female | 32 (69.6) |
| Missing | 5 (10.9) |
| **Ethnicity, n (%)** | |
| White | 39 (84.8) |
| Black | 1 (2.2) |
| Prefer not to say | 1 (2.2) |
| Missing | 5 (10.9) |
| **Age in years, n (%)** | |
| <40 | 5 (10.9) |
| 40–49 | 10 (21.7) |
| 50–59 | 15 (32.6) |
| 60–69 | 7 (15.2) |
| 70–79 | 3 (6.5) |
| Missing | 6 (13.0) |
| **Site, n (%)** | |
| 1 (South East England) | 8 (17.4) |
| 2 (South East England) | 7 (15.2) |
| 3 (South West England) | 15 (32.6) |
| 4 (North West England) | 10 (21.7) |
| 5 (South West England) | 6 (13.0) |

FREE-IA, Fatigue - Reducing its Effects through individualised support Episodes in Inflammatory Arthritis.

one of the new sites who had been unable to attend the group session with colleagues.

## Patient recruitment and intervention delivery

A total of 46 patients were recruited to the FREE-IA study (figure 1, table 2). The overall recruitment rate was 0.22 participants per hospital per month; however, most sites did not recruit continuously over the duration of the recruitment period. The conversion rate, based on the number of participants recruited divided by the number screened, was 52.1% (63/121). Six of the 63 patients (9.5%) who expressed interest in participating were ineligible and/or declined to participate. Of the remaining 57 patients, five did not provide consent (8.8%) and three declined an invitation to take part (5.3%). One site did not invite an eligible patient because they had reached their target recruitment and one site stopped recruitment early due to COVID-19, with the local team unable to invite two interested and eligible patients to participate in the study. This left 46 patients who provided written consent and who provided a proposed primary outcome at baseline.

Eight RHPs delivered 113 intervention sessions across five sites and duration ranged from 10 to 120 min (mean 44 min). One RHP took consent but did not deliver the intervention. At two sites, all intervention sessions were delivered by one RHP. At the three other sites, the number of intervention sessions delivered by each RHP varied. Of the total 46 participants, 39 (84.8%) completed the two core sessions. Seven (15.2%) attended one session, 16 (34.8%) attended two sessions, 18 (39.1%) attended three sessions and 5 (10.9%) attended the maximum four possible sessions. Mode of delivery was face to face, except for four optional sessions, which were delivered by telephone. Session 2 of the intervention was delivered within the desired 2 week timeframe for 37% of the participants who attended at least the two core sessions, with a mean of 21 days between sessions. No adverse events were reported.

Twenty-five intervention sessions were audio-recorded across three sites; two sites did not record any sessions. AB evaluated all the audio-recordings, and SB and ED analysed a subset independently. There was a high level of agreement between the team members in relation to the audio-recordings that were analysed in triplicate. The main insights were that:

► Most RHPs followed the manual in a highly structured, linear way, but some adopted a more flexible approach guided by patients' fatigue-related support needs.
► RHPs used the materials to prompt discussion using a non-didactic approach, initially to explore fatigue drivers and daily diaries, and later to explore goal setting, sleep and stress.
► When it was difficult for patients to identify unhelpful behaviour patterns, some RHPs were more directive.
► Longer appointments allowed for linking thoughts and feelings with behaviours, developing goals and exploring behaviour patterns.
► RHPs who had more time and/or experience and/or knew the patient from previous clinical appointments tended to explore negativity towards change with more confidence.

## Data completeness and summary of patient-reported outcome measures

There were 13 (9.4%) missing proposed primary outcome responses from 11 participants (T0=0, T1=6, T2=8) and 27 (19.6%) missing secondary outcome responses from 18 participants (T0=6, T1=12, T2=11). This meant that 87% of participants completed the proposed primary outcome measure postintervention and 82.6% of participants completed the proposed primary outcome measure at 6 months (figure 1). The completeness of each of the outcome measures was also high (online supplemental table S1).

Summary statistics of each (sub)score across time are shown in table 3. Results indicated improvement in all measures at either T1 or T2, or both except for disability (table 4). Improvements in the fatigue measures were in line with published clinically meaningful changes.[37]

Results from the health economic analysis are presented in table 4. The key cost driver for this patient group was

**Table 3** Summary of participant-reported outcome measures at all time points and mean differences with corresponding 95% CIs

| Measure (scale range) | T0 Mean (SD) (range) | T1 Mean (SD) (range) | T2 Mean (SD) (range) | T1-T0 Mean difference (95% CI) | T2-T0 Mean difference (95% CI) |
|---|---|---|---|---|---|
| BRAF-NRS fatigue effect (0–10) | 8.48 (1.19) (6.00–10.00) (n=46) | 6.68 (1.54) (4.00–9.00) (n=40) | 6.03 (2.72) (0.00–10.00) (n=39) | −1.78 (−2.27 to −1.28) (n=40) | −2.41 (−3.29 to −1.53) (n=39) |
| BRAF-NRS coping (0–10) | 6.68 (2.25) (1.00–10.00) (n=41) | 5.79 (2.53) (0.00–10.00) (n=34) | 5.03 (2.72) (0.00–10.00) (n=34) | −0.59 (−1.53 to 0.34) (n=32) | −1.06 (−2.00 to −0.12) (n=32) |
| RAID final score (0–10) | 6.40 (1.60) (1.87–9.25) (n=41) | 5.57 (2.00) (1.65–8.79) (n=34) | 5.54 (1.91) (1.30–8.79) (n=36) | −0.64 (−1.27 to −0.00) (n=32) | −0.61 (−1.32 to 0.10) (n=33) |
| BRAF-MDQ physical severity (0–22) | 17.92 (2.82) (11.00–22.00) (n=41) | 14.97 (4.16) (5.00–22.00) (n=34) | 14.56 (5.22) (4.00–22.00) (n=34) | −2.44 (−3.75 to −1.12) (n=32) | −2.87 (−4.85 to −0.89) (n=30) |
| BRAF-MDQ living with fatigue (0–21) | 12.42 (4.95) (4.00–21.00) (n=41) | 9.09 (6.10) (0.00–21.00) (n=34) | 8.63 (5.88) (0.00–21.00) (n=34) | −2.75 (−4.52 to −0.98) (n=32) | −2.72 (−4.55 to −0.88) (n=30) |
| BRAF-MDQ cognitive (0–15) | 9.39 (3.93) (1.00–15.00) (n=41) | 7.62 (3.82) (0.00–15.00) (n=34) | 7.09 (3.51) (1.00–15.00) (n=34) | −1.84 (−3.19 to −0.50) (n=32) | −1.63 (−3.22 to 0.05) (n=30) |
| BRAF-MDQ emotional (0–12) | 7.71 (3.16) (1.00–12.00) (n=41) | 5.44 (3.51) (1.00–12.00) (n=34) | 5.47 (3.52) (0.00–12.00) (n=34) | −1.47 (−2.51 to −0.42) (n=32) | −1.67 (−3.06 to −0.27) (n=30) |
| BRAF-MDQ total (0–70) | 47.43 (12.60) (21.00–66.00) (n=41) | 37.12 (15.39) (14.00–68.00) (n=34) | 35.75 (15.84) (9.00–66.00) (n=34) | −8.50 (−13.03 to −3.97) (n=32) | −8.88 (−15.00 to −2.77) (n=30) |
| MHAQ mean score (0–4) | 0.84 (0.58) (0.00–2.38) (n=41) | 0.72 (0.55) (0.00–2.13) (n=33) | 0.81 (0.61) (0.00–2.00) (n=34) | −0.07 (−0.23 to 0.08) (n=31) | 0.03 (−0.15 to 0.21) (n=31) |
| HCCQ (1–7)* | 3.95 (1.50) (1.17–7.00) (n=39) | 5.46 (1.36) (2.00–7.00) (n=34) | 4.85 (1.69) (1.33–7.00) (n=36) | 1.35 (0.65 to 2.05) (n=31) | 1.01 (0.35 to 1.67) (n=32) |
| RASE (28–140)* | 100.16 (12.20) (78.00–128.00) (n=38) | 105.67 (13.36) (72.00–140.00) (n=33) | 104.32 (16.21) (72.00–135.00) (n=35) | 3.32 (−0.62 to 7.26) (n=31) | 4.80 (1.00 to 8.60) (n=32) |

*Higher scores indicate better outcome.
BRAF-MDQ, BRAF Multidimensional Questionnaire; HCCQ, Health Care Climate Questionnaire; MHAQ, Modified Health Assessment Questionnaire; RAID, Rheumatoid Arthritis Impact of Disease; RASE, Rheumatoid Arthritis Self-Efficacy Scale.

medication use, with very costly biologicals driving the overall medication costs for some participants. Other substantial contributors to the overall cost from the NHS/PSS perspective were hospital inpatient, outpatient and day cases. Care costs (both informal and privately paid) represented considerable cost burdens from the patient perspective. The mean delivery cost was estimated to be £98.40 per participant, rising to £128 when training costs were included.

## DISCUSSION

During the FREE-IA study, RHPs delivered over 100 intervention sessions to patients struggling with the impact of fatigue. Results from the participant-reported outcomes suggest that this flexible, low-cost intervention has the potential to help patients self-manage this symptom. There is existing evidence for the effectiveness of higher

intensity interventions delivered over several weeks to groups of patients.[18 19] If the fatigue-related support needs of some patients could be met with a lower intensity intervention delivered over fewer sessions, it could increase choice and provision. The evidence that RHPs from different professional backgrounds undertook training and delivered the intervention further increases the possibility that this type of support could be practical to provide in a range of clinical settings. Although some sessions lasted for longer than the guideline of 20–30 min, most participants did not take up the maximum four sessions, with half attending three sessions and around 10% attending all four sessions. The intervention was estimated to be delivered at a relatively low cost per participant. Although the FREE-IA study sample is too small to evaluate whether duration and number of intervention sessions influenced outcomes, results suggest that two to

**Table 4** Economic evaluation measures: resource use, costs and outcomes over 6 months of follow-up

| Resource use | n | Mean resource use per participant (SD) | Mean costs per participant (£) | 95% CI |
|---|---|---|---|---|
| A&E visits | 35 | 0.14 (0.36) | 23.71 (58.94) | |
| Outpatient visits | 30 | 1.43 (1.76) | 210.70 (258.05) | |
| Day cases | 30 | 0.40 (1.33) | 300.80 (999.20) | |
| Inpatient stays | 30 | 0.10 (0.31) | 224.57 (777.42) | |
| GP appointments | 34 | 1.94 (2.37) | 66.00 (80.69) | |
| Nurse appointments | 34 | 1.56 (2.26) | 16.91 (24.51) | |
| GP home visits | 30 | 0 (0) | 0.00 (0.00) | |
| Nurse home visits | 30 | 0.07 (0.37) | 1.47 (8.05) | |
| Medications | 30 | 2.57 (1.41) | 2729.66 (2796.45) | |
| Nurse helpline | 35 | 0.66 (1.03) | 37.13 (58.05) | |
| Carer contacts | 35 | 5.94 (30.95) | 68.34 (355.90) | |
| Total cost (NHS/PSS perspective) | | | 3690.08 (3660.83) | 2323.10 to 5057.05 |
| Informal care contacts | 35 | 71.33 (165.20) | 621.99 (1440.58) | |
| Private healthcare | | | 82.33 (180.38) | |
| Private carers | | | 128.03 (365.83) | |
| Total cost (patient perspective) | | | 624.83 (1072.68) | 224.28 to 1025.37 |
| Outcomes | n | Mean QALYs | | |
| QALYs over the 6-month period | 27 | 0.275 (0.105) | | 0.23 to 0.32 |

A&E, Accident & Emergency; GP, general practitioner ; n, all available data were used for each type of resource use or outcome; NHS, National Health Service; PSS, personal social service; QALYs, quality-adjusted life years.

three sessions might be enough for patients to derive clinically meaningful benefit.

An appropriate next step is to design and conduct a definitive RCT to test the clinical and cost-effectiveness of our intervention. This single-arm feasibility study explored several uncertainties and has provided insights to inform the design and delivery of such a study. These include understanding variation in local processes and the resources available to support recruitment and intervention delivery, for example, how to identify and invite potential participants and how to collect consent with minimal impact on the workload and time of RHPs. Collecting the proposed primary outcome by telephone and secondary outcomes via mail was a successful strategy overall. However, it was not always possible to contact participants by telephone or convenient for them to

respond at that time. Returning paper outcomes in the mail might have been difficult, for example, due to 'shielding' during the COVID-19 pandemic (namely, people who were advised not to leave their homes and to minimise all face-to-face contact). In a future study, we would seek ethics approval to incorporate options to contact participants by text and email and to collect outcomes online, as well as including the telephone and paper options. Improvements to the Resource Use Questionnaire were identified, allowing an optimised approach for a definitive RCT. The small number of audio-recorded sessions suggests that we need to find a different approach to evaluating competency and fidelity. Anecdotal feedback from RHPs suggests that gaining consent for audio-recording at the start of the intervention session took up too much time and audio-recording altered the interaction with participants, making it less like 'real life' clinical practice. We also need to reconsider the aim to deliver core session 2 within 2 weeks of core session 1, given that RHPs and/ or patients were often unable to do this. Reasons for this were not systematically captured but included difficulty booking and/or attending clinic appointments within the short timeframe. A key rationale for this timeframe was to review participants' activity diaries, one of the intervention tools introduced in session 1 (table 1). Options in the future include providing activity diaries to cover a longer period or having brief activity diary reviews by telephone between intervention sessions.

While our results suggest that a definitive RCT is feasible and our intervention has the potential to be helpful to patients, the large-scale changes in rheumatology care provision in response to the COVID-19 pandemic will impact the next steps.[38 39] The move from face to face to telephone and video consultations is likely to result in long-term changes and has implications for the testing and possible implementation of our intervention. However, the clear and careful design of FREE-IA mean that the training and intervention are well positioned to be adapted for delivery in a range of modes and settings, including online. Although remote delivery of sessions was barely used in the current study, many patients and RHPs are becoming more familiar and comfortable with telephone and/or video interactions.[40 41] In addition to influencing current practice, aspects of the intervention could inform professional preregistration education programmes therefore helping another generation of NHS health professionals to support patients to self-manage their fatigue.

Study strengths include the low levels of attrition and the high levels of completed outcomes collected. Standardised outcome collection was ensured by the central team who were external to the hospitals delivering the intervention. As well as informing the design of a definitive RCT, our flexible, pragmatic approach to local variation meant that we gained insights into how the intervention could be delivered in clinical practice. This study benefited from the input of two patient research partners, MU and BA, who contributed throughout the

study, from identifying the research question through to interpreting the results. Feedback from the Patient Advisory Group of the Rheumatology Department at the Bristol Royal Infirmary also enhanced the study.

Study limitations include the lack of a control arm. To maximise information relating to the intervention itself, and given limited resources, we did not include a concurrent control group and hence have not tested the feasibility/acceptability of randomisation. However, given that the intervention is not available in routine care, it is likely that patients willing to try the intervention, as in this study, are also likely to accept randomisation. This was a feasibility study and as such the data on health-related outcomes should not be overinterpreted: the improvements seen are within-patient comparisons only, hence could arise from regression to the mean or the small sample size. However, outcomes were in the direction to suggest the intervention could have a beneficial impact on patients' fatigue, and CIs support an interpretation of improvement. Our proposed primary outcome is the BRAF-NRS Fatigue Effect, which was developed with patients who have RA, although it has subsequently been validated in patients with psoriatic arthritis.[42] There might also be other important outcomes, such as work productivity, that we could include in a future trial.

## Conclusions

We were able to design and deliver intervention training to RHPs, who were then able to deliver intervention sessions to participants, guided by the intervention manual. However, it was not always possible to deliver core session 2 within the desired 2-week timeframe. We were able to collect outcomes at three time points and had low levels of attrition. Overall, our results suggest that a definitive RCT is feasible. While being cautious, outcomes were in a direction to suggest improvement in participants' fatigue impact after attending relatively low-cost intervention sessions.

**Author affiliations**
[1]Academic Rheumatology, Bristol Royal Infirmary, Bristol, UK
[2]School of Health and Social Wellbeing, University of the West of England, Bristol, UK
[3]Health Sciences, University of Southampton, Southampton, UK
[4]Division of Clinical Psychology, Uppsala University, Uppsala, Sweden
[5]College of Medicine and Health, University of Exeter, Exeter, UK
[6]Peninsula Medical School, Plymouth University, Plymouth, UK
[7]Bristol Population Health Science Institute, University of Bristol, Bristol, UK
[8]Taunton and Somerset NHS Foundation Trust, Taunton, UK

**Contributors** ED: funding acquisition, study conceptualisation, methodology, study supervision and writing original draft. SB: data collection, study management, data analysis and reviewed draft. BA and LMM: study conceptualisation, study delivery and reviewed draft. JA: study conceptualisation and reviewed draft. AB: data collection, data analysis and reviewed draft. SC: methodology, study delivery, formal data analysis and reviewed draft. SH: study conceptualisation, methodology, study delivery and reviewed draft. JL: formal data analysis and reviewed draft. MN: methodology and reviewed draft. JT and PE: methodology, formal data analysis and reviewed draft. MU: study conceptualisation, study delivery and reviewed draft. ED is the guarantor.

**Funding** This work was supported by a National Institute for Health Research (NIHR) grant under its research for Patient Benefit (RfPB) Programme (Grant Reference Number PB-PG-1216-20014). This publication presents independent research funded by the NIHR.

**Disclaimer** The views expressed are those of the authors and not necessarily those of the NHS, the NIHR or the Department of Health and Social Care.

**Competing interests** None declared.

**Patient and public involvement** Patients and/or the public were involved in the design, or conduct, or reporting, or dissemination plans of this research. Refer to the Methods section for further details.

**Patient consent for publication** Not applicable.

**Ethics approval** Ethics approvals for the study were granted by the South West – Frenchay Research Ethics (REC ref. 15/SW/0207). All participants provided written informed consent prior to taking part in the study.

**Provenance and peer review** Not commissioned; externally peer reviewed.

**Data availability statement** Data are available upon reasonable request. All data relevant to the study are included in the article or uploaded as supplementary information. Data will be available from the lead author on request.

**ORCID iDs**
Emma Dures http://orcid.org/0000-0002-6674-8607
Jo Adams http://orcid.org/0000-0003-1765-7060
Sarah Hewlett http://orcid.org/0000-0001-7851-2039
Joanna Thorn http://orcid.org/0000-0001-8962-2428

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
