## [Reviewer comments · BMJ Open]

ARTICLE DETAILS

TITLE (PROVISIONAL)	A brief intervention to reduce fatigue impact in patients with inflammatory arthritis: design and outcomes of a single-arm feasibility study
AUTHORS	Dures, Emma; Bridgewater, Susan; Abbott, Bryan; Adams, Jo; Berry, Alice; McCracken, Lance; Creanor, Siobhan; Hewlett, Sarah; Lomax, Joe; Ndosu, Mwidimi; Thorn, Joanna; Urban, Marie; Ewings, Paul

VERSION 1 – REVIEW

REVIEWER	Stebbing, Simon Otago University, Medicine
REVIEW RETURNED	08-Sep-2021

GENERAL COMMENTS	This is a timely and interesting paper, and the authors are to be commended for taking a systematic approach to develop a fatigue intervention for patients with inflammatory arthritis and assess the feasibility of this intervention including associated costs. A significant amount of work has gone into the study, but unfortunately there are some major issues with the reporting of the methods, and in particular the intervention. Major revision is required and I would hope that missing information in the paper is available for analysis and or presentation in order for the paper to be published. Major 1. Definition of the study population. The study participants are stated as having 'inflammatory arthritis' and entry criteria were: 'a clinician-confirmed diagnosis of IA'. This is a very broad term and a very loose categorization for inclusion. This poses significant issues; 1. The term IA in the introduction is used interchangeably with rheumatoid arthritis (RA). 2. Most research into fatigue has been undertaken in patients with a defined RA diagnosis. 3. The BRAF – the primary outcome measure is a measure of fatigue validated for RA not for 'IA' 4. IA could include conditions with very different demographics and clinical presentations and influences on fatigue – such as Ankylosing spondylitis. There is no demographic breakdown of sub-diagnosed or clinical features of the IA group. 2. Disease activity. Similarly, we are told in the introduction that in patients with IA, fatigue is affected by disease activity and disability. Whilst disability is assessed by MHAQ, there is no measure of disease activity and no documentation of drug therapy. So how do we know that changes in fatigue were not influenced by changes in disease activity or drug therapy during the course of the study?
--

	3. Fatigue is a variable symptom. This should be further explored in the introduction. Studies show variation over time and even over the course of a day. How did the study control for natural variation in fatigue? For instance at different times of the day when consultations took place , or other life events supervened. Controlling for all these may be impossible, but the BRAF looks at average fatigue over one week. This should be mentioned and discussed. 4. The QALYs and economic impact is reported but there is a major opportunity lost to evaluate presenteeism and absenteeism due to fatigue and any impact that the intervention may have on this- including time for appointments. In the introduction the authors make the point that 75% of patients with IA are of working age, so surely work productivity is a seriously important outcome? As this is not addressed it should be incorporated into the discussion as a weakness of the study. 5. As a single arm feasibility study there is no problem with not having a control group, but the purpose of a feasibility study is partly to address how a future full scale randomised trial should be conducted. This should include the choice of primary outcome, the minimum clinically meaningful benefit for this outcome, the likelihood that the intervention will achieve this, how many patients are likely to be needed in a full sized study, with an estimated a priori power analysis based on this. Furthermore, a comment on the best control intervention is needed in a future randomised trial. What options are there for control interventions in a complex intervention of this sort- eg usual care, patient education, within person (wait list) or cross over controls. 6. A frustrating aspect of this study is that there is no clear description of the intervention, other than in table 1, and no description of how the intervention was developed. The authors cite the Medical Research Council's framework for developing and evaluating complex interventions. As part of this process a reader should be able to reproduce the study for themselves. Information regarding the intervention is very sketchy and what is missing is a report on how the intervention was developed. This could be a paper in itself that could be referenced. At minimum the manual given to the Rheumatology health care professionals should provided as an appendix online. I would like to understand how aspects of the intervention were chosen and the background to this. Were patients consulted on the content of the intervention. If so, was this part of a qualitative or thematic analysis? The paper concentrates on the implementation of the intervention with very little about the intervention itself. Minor:  1. The introduction should discuss variability in fatigue and its measurement. The need to use measures of quantity, impact and coping and the best measuring instruments 2. Page 7 line 12. The aims of the study should be set out in a new paragraph 3. Page 8 line 22. In the discussion a comment should be made regarding potential for selection bias for patients entering the study. 4. Page 9 line 44-51 The questionnaires: The Rheumatoid Arthritis Self-Efficacy Scale (RASE), The Perceived Health Competence Scale (PHCS), The Health Care Climate Questionnaire (HCCQ) all need to be described and the purpose for which they are designed and used in the study explained. How are they scored ? how many items do they have etc.
--	--

	5. P10 line 49 'The research study, including the question, was developed with patient research partners' . This doesn't make sense, perhaps this could be clarified? 6. Table 4 the mean difference between the time points measured is difficult to interpret, adding a percentage change would be helpful 7. Table 5 Title of this table and column headings need to be revised as they are not clear
--	---

REVIEWER	O'Meara, Susan Kleijnen Systematic Reviews Ltd
REVIEW RETURNED	27-Jan-2022

GENERAL COMMENTS	Thank you for submitting your interesting manuscript describing a single-arm feasibility study assessing a brief intervention to reduce the impact of fatigue in patients with inflammatory arthritis. I have assessed the study using the CONSORT 2010 checklist for reporting a pilot or feasibility trial. The study report is very well-written and includes a good level of detail describing the context, rationale, methods, findings and ambitions to undertake a full clinical trial and cost-effectiveness analysis in the future. The vast majority of the items on the CONSORT checklist have been covered in the paper and therefore my further points of feedback are minor: please find these below. Abstract Objectives: I suggest making it a little clearer that 'FREE-IA' is the name of the study Methods: 'RHPs delivered 2-4 sessions.....' Please state over what time period this took place. Conclusions: maybe word this a little more cautiously e.g., '.....the observed within-group improvements suggest that it may be worthwhile to assess the clinical effectiveness of the intervention within a clinical trial' (or similar). Strengths and limitations of this study (bullet points) 3rd and 4th points seem to cancel each other out. As above, I suggest that you word the conclusions (3rd point) with more caution. Main text Page 8/28 and onwards: although you include some useful details in Table 1, it's not entirely clear how the health professionals delivered the intervention. For example, was each session highly structured or informal/exploratory? Was it led by the health professional or the patient or was this shared? Was it in the form of counselling or therapy or more like an educational session? Page 9/28, line 12: what approach did the research fellow use to score the content of the intervention? Page 11/28: there are some useful details here about the health professionals but perhaps additional details could go in a table e.g., number of years qualified, level of experience with delivering this type of intervention, level of experience with supporting patients with self-management activities. Tables The series of data tables are well-detailed and clearly set out. Tables 3 and 4 are particularly well-presented with sufficient detail to aid interpretation of the results. To complement these, I'm
--

	wondering if the authors have thought about assessing unintended harms of the intervention for both providers and participants. This may be something to think about for the full clinical trial. Finally, I did not see mention of a study protocol and I wondered if this could be made available for readers. I hope that my comments are helpful and I wish you all the best in progressing this research to a fully-powered clinical trial.
--	---

VERSION 1 – AUTHOR RESPONSE

Response to reviewers

We thank the reviewers for their detailed and considered feedback. We have addressed their comments below together with suggested changes to the manuscript.

Definition of the study population. The study participants are stated as having ‘inflammatory arthritis’ and entry criteria were: ‘a clinician-confirmed diagnosis of IA’. This is a very broad term and a very loose categorization for inclusion. This poses significant issues;

1. *The term IA in the introduction is used interchangeably with rheumatoid arthritis (RA).*
 - We recognise that rheumatoid arthritis (RA) is the most common type of inflammatory arthritis (IA). Our intention in using both “RA” and “IA” in the introduction was to specify whether we were citing RA-specific studies or studies that also sampled for other types of IA. We have added a sentence on page 4 to name some of these other IA conditions and to clarify that they are all managed in rheumatology services.
2. *Most research into fatigue has been undertaken in patients with a defined RA diagnosis.*
 - We agree that to date, most of the research into fatigue has been conducted with patients with a diagnosis of RA. However, the evidence is growing that fatigue is also a common and challenging symptom for patients with other types of IA. We agree this is an important point to include and have added in a sentence on page 4 to highlight this and cited two recent reviews.
3. *The BRAF – the primary outcome measure is a measure of fatigue validated for RA not for ‘IA’*
 - We acknowledge that the BRAF was developed and validated with patients with RA, and we have now added this as a limitation on page 17. We have also added in that the BRAFs have subsequently been validated in patients with psoriatic arthritis.
4. *IA could include conditions with very different demographics and clinical presentations and influences on fatigue – such as ankylosing spondylitis. There is no demographic breakdown of sub-diagnosed or clinical features of the IA group.*
 - We appreciate this important point about diagnoses and clinical features. We did not collect this data because our intervention is based on a psychologically informed ‘transdiagnostic’ approach. We looked at the feasibility of a self-management intervention that is designed to address individual health behaviours and coping skills across conditions, rather than resolve or ‘cure’ fatigue.

Disease activity. Similarly, we are told in the introduction that in patients with IA, fatigue is affected by disease activity and disability. Whilst disability is assessed by MHAQ, there is no measure of disease activity and no documentation of drug therapy. So how do we know that changes in fatigue were not influenced by changes in disease activity or drug therapy during the course of the study?

- We did not intend to imply that there is a straightforward association between fatigue and disease activity and agree this can be made clearer for the readers. We have re-written parts of the introduction on page 4 to clarify this. We wanted to convey that although some studies have shown a correlation, the evidence is still far from conclusive. It is possible that the causal and maintaining factors of fatigue vary both within and between people. The limited

resources available for this feasibility study meant that we had to focus on collecting data pertinent to addressing the key uncertainties of a future trial. We agree with the reviewer that with this feasibility design, we cannot attribute changes in fatigue severity to the effects of the intervention and we have clarified in the text to confirm this was never an aim of this single-arm feasibility study.

Fatigue is a variable symptom. This should be further explored in the introduction. Studies show variation over time and even over the course of a day. How did the study control for natural variation in fatigue? For instance, at different times of the day when consultations took place, or other life events supervened. Controlling for all these may be impossible, but the BRAF looks at average fatigue over one week. This should be mentioned and discussed.

- Thank you for this observation. We have now added in details about the variable and fluctuating nature of fatigue on page 4. We agree that it would be close to impossible to control for the natural variation in fatigue severity in a definitive study design. However, a definitive trial would likely be randomised, enabling us to compare outcomes in the control and intervention arms. Our proposed primary outcome is fatigue effect, and it is possible that the impact of the symptom does not vary in the same way as fatigue severity. On page 5, we explain that this is because we theorise that impact is not only influenced by the symptom but also by the meaning that the person makes of it and their physical, cognitive and emotional responses.

The QALYs and economic impact is reported but there is a major opportunity lost to evaluate presenteeism and absenteeism due to fatigue and any impact that the intervention may have on this- including time for appointments. In the introduction the authors make the point that 75% of patients with IA are of working age, so surely work productivity is a seriously important outcome? As this is not addressed it should be incorporated into the discussion as a weakness of the study.

- We thank the reviewer for drawing attention to this issue. On page 17, we have added it in as a limitation of the current study and something that we will consider in future work.

As a single arm feasibility study there is no problem with not having a control group, but the purpose of a feasibility study is partly to address how a future full scale randomised trial should be conducted. This should include the choice of primary outcome, the minimum clinically meaningful benefit for this outcome, the likelihood that the intervention will achieve this, how many patients are likely to be needed in a full-sized study, with an estimated a priori power analysis based on this. Furthermore, a comment on the best control intervention is needed in a future randomised trial. What options are there for control interventions in a complex intervention of this sort- eg usual care, patient education, within person (wait list) or cross over controls.

- We agree that these are important points which will be addressed in the design of a future study. We anticipate that any follow-on definitive trial would be of a pragmatic nature, and hence the control group would be usual care/treatment as usual – with the caveat that given the pandemic, and longer-term implications for changes in service delivery, at this time it is difficult to define/describe treatment as usual.

A frustrating aspect of this study is that there is no clear description of the intervention, other than in table 1, and no description of how the intervention was developed. The authors cite the Medical Research Council's framework for developing and evaluating complex interventions. As part of this process a reader should be able to reproduce the study for themselves. Information regarding the intervention is very sketchy and what is missing is a report on how the intervention was developed. This could be a paper in itself that could be referenced. At minimum the manual given to the Rheumatology health care professionals should provided as an appendix online.

- We appreciate the reviewer's frustration about the lack of detail. We have added in some more detail on page 5. In addition, we have provided a single page summary describing the intervention. We conducted a parallel qualitative process evaluation as part of the FREE-IA study, which will also be submitted for publication. This qualitative process evaluation manuscript will also provide further information. At present, we do not think it is appropriate to provide the manual as an appendix because it is designed to be used by health professionals after they have taken part in training, rather than as a stand-alone tool. Also, as this is a feasibility study, we have not tested whether the manual is effective. We aim to publish this in a definitive trial when we have conducted a fully powered analysis of the effectiveness of the training and manual.

I would like to understand how aspects of the intervention were chosen and the background to this. Were patients consulted on the content of the intervention. If so, was this part of a qualitative or thematic analysis? The paper concentrates on the implementation of the intervention with very little about the intervention itself.

- In this paper we have focused on the feasibility of delivering the intervention and collecting outcomes, rather than the intervention development. We can clarify that the intervention was informed by theories of self-efficacy and self-determination and included the use of cognitive-behavioural approaches and Socratic questioning (see page 5 and the single-page intervention summary). The structure and content were informed by telephone interviews with UK rheumatology clinicians about what they could deliver in clinical practice, observations and feedback from a fatigue clinic and selected materials from a tested fatigue self-management intervention delivered to groups. Patients were collaborators throughout this process, including during intervention development.

Minor:

1. The introduction should discuss variability in fatigue and its measurement. The need to use measures of quantity, impact and coping and the best measuring instruments

- We have added in information about fatigue variability and the challenges of measurement on page 4.

2. Page 7 line 12. The aims of the study should be set out in a new paragraph

- Thank you, we have done this.

3. Page 8 line 22. In the discussion a comment should be made regarding potential for selection bias for patients entering the study.

- We appreciate the importance of selection bias. We do not have any reason to believe that there was potential for selection bias in this study, beyond the generic challenges in any trial. However, we will revisit the issue when designing the future definitive RCT.

4. Page 9 line 44-51

The questionnaires: The Rheumatoid Arthritis Self-Efficacy Scale (RASE), The Perceived Health Competence Scale (PHCS), The Health Care Climate Questionnaire (HCCQ) all need to be described and the purpose for which they are designed and used in the study explained. How are they scored? how many items do they have etc.

- We have added in the purpose of the questionnaires on page 9. Information on how they were scored is included in Table 3.

5. P10 line 49 'The research study, including the question, was developed with patient research partners'. This doesn't make sense, perhaps this could be clarified?

- On page 10, we have clarified that the patient research partners helped us to develop the specific study aims.

5. Table 4 the mean difference between the time points measured is difficult to interpret, adding a percentage change would be helpful

We have combined Table 3 and Table 4, and the revised table shows summary measures at all time points and the corresponding mean differences, making it easier for the reader to follow.

7. Table 5 Title of this table and column headings need to be revised as they are not clear

- We have amended the title and column headings in Table 5 as suggested to improve the clarity.

Reviewer: 2

Dr. Susan O'Meara, Kleijnen Systematic Reviews Ltd Comments to the Author:

Thank you for submitting your interesting manuscript describing a single-arm feasibility study assessing a brief intervention to reduce the impact of fatigue in patients with inflammatory arthritis.

I have assessed the study using the CONSORT 2010 checklist for reporting a pilot or feasibility trial. The study report is very well-written and includes a good level of detail describing the context,

rationale, methods, findings and ambitions to undertake a full clinical trial and cost-effectiveness analysis in the future. The vast majority of the items on the CONSORT checklist have been covered in the paper and therefore my further points of feedback are minor: please find these below.

Abstract

Objectives: I suggest making it a little clearer that 'FREE-IA' is the name of the study

- Thank you, we have clarified that FREE-IA is the study name.

Methods: 'RHPs delivered 2-4 sessions.....' Please state over what time period this took place.

- We are at the maximum word limit in the abstract. However, we appreciate the importance of the information, which we have included on page 12. If there is scope to go over the abstract word limit a little, we would of course be happy to include this information.

Conclusions: maybe word this a little more cautiously e.g., '.....the observed within-group improvements suggest that it may be worthwhile to assess the clinical effectiveness of the intervention within a clinical trial' (or similar).

- Thank you for the suggestion – we have amended the text to reflect your point.

Strengths and limitations of this study (bullet points) 3rd and 4th points seem to cancel each other out. As above, I suggest that you word the conclusions (3rd point) with more caution.

- We have amended bullet points 3 & 4 to be more cautious overall.

Main text

Page 8/28 and onwards: although you include some useful details in Table 1, it's not entirely clear how the health professionals delivered the intervention. For example, was each session highly structured or informal/exploratory? Was it led by the health professional or the patient or was this shared? Was it in the form of counselling or therapy or more like an educational session?

- We agree that these are important points, with implications for fidelity and competency. As part of the FREE-IA study, we conducted a parallel qualitative process evaluation. We are currently finalising a second manuscript to report this process evaluation, which provides further details about how the health professionals delivered the intervention. In the 'Analysis and results' section on page 12 of this manuscript, we have clarified that there was individual variation in the extent to which health professionals delivered the sessions in a highly structured, linear way based closely on the manual (compared to being more loosely guided by the manual in response to the patients' input). However, we can confirm that all the health professionals used a non-didactic approach to deliver the intervention. We have also included a single page summary of the intervention, which addresses some of these points, as a supplement.

Page 9/28, line 12: what approach did the research fellow use to score the content of the intervention?

- On page 8, we have added detail about the approach used by the research fellow to score the content of the intervention.

Page 11/28: there are some useful details here about the health professionals but perhaps additional details could go in a table e.g., number of years qualified, level of experience with delivering this type of intervention, level of experience with supporting patients with self-management activities.

- We did not collect this information for several reasons, including the difficulty of ensuring anonymity (with only a small number of health professionals involved in this feasibility study) and the challenge of defining and assessing relevant experience (e.g., one health professional had limited experience of rheumatology but had worked in chronic fatigue previously). This is something that we can consider when designing a future study,

Tables

The series of data tables are well-detailed and clearly set out. Tables 3 and 4 are particularly well-presented with sufficient detail to aid interpretation of the results. To complement these, I'm wondering if the authors have thought about assessing unintended harms of the intervention for both providers and participants. This may be something to think about for the full clinical trial.

- It was a requirement of our sponsor that we had systems in place to record, investigate and report adverse events arising from our research. We wrote the 'FREE-IA – Guidelines for Safety Reporting – May 2019' which we sent to all participating sites. We agree that this is important information and on page 12, we have confirmed that no adverse events were reported during the study.

Finally, I did not see mention of a study protocol and I wondered if this could be made available for readers.

- We did not publish our protocol as our study was a non-randomised feasibility design. However, we are happy to provide a copy of the protocol on request or to include it as an appendix/supplementary information if the Editor would like this.

I hope that my comments are helpful and I wish you all the best in progressing this research to a fully-powered clinical trial.

- Thank you for your supportive feedback.

VERSION 2 – REVIEW

REVIEWER	O'Meara, Susan Kleijnen Systematic Reviews Ltd
REVIEW RETURNED	31-Mar-2022
GENERAL COMMENTS	I am satisfied that the manuscript authors have addressed all comments.